# Influence of Natural Weather Conditions in the Long-Term Fracture Energy of Glass Fibre Reinforced Cement (GRC) Modified with Chemical Additions

**DOI:** 10.3390/ma14123355

**Published:** 2021-06-17

**Authors:** Alejandro Enfedaque, Marcos G. Alberti, Jaime C. Gálvez, Shou Mengie

**Affiliations:** 1Departamento de Ingeniería Civil: Construcción, E.T.S de Ingenieros de Caminos, Canales y Puertos, Universidad Politécnica de Madrid, 28040 Madrid, Spain; marcos.garcia@upm.es (M.G.A.); jaime.galvez@upm.es (J.C.G.); 2Department of Civil Engineering, University of Tongji, Shanghai 200092, China; smj92jessie@gmail.com

**Keywords:** GRC, glass fibres, cement mortar, fracture energy, aging methods, digital image correlation

## Abstract

The use of glass fibre-reinforced cement (GRC) in structural elements has been limited due to the reduction in the mechanical properties of the material with aging. Chemical additions have been used to modify the cement mortar formulation in order to minimise such loss, but no conclusive results have been obtained yet. Moreover, the application of accelerated aging methods in such modified GRC formulations still poses several uncertainties. An experimental campaign seeking to assess the reduction in the fracture energy of two GRCs manufactured with modified matrixes after five years of exposure to natural environment was performed. Furthermore, a comparison with results from the literature that used accelerated aging methods was performed. The results show that the use of the chemical additives might be capable of maintaining to a notable extent the mechanical properties of GRC after five years of natural aging. Regarding the accelerated aging method by means of immersion in hot water tanks, it seemed that the equivalences applied in previous research accurately match the degradation of the material after natural exposure to weather. Additionally, a digital image correlation analysis showed that aged GRCs seemed to distribute damage in a smaller area than young GRCs.

## 1. Introduction

Since 1950, fibres have been profusely used in the reinforcement of concrete and cement mortars because they can improve several properties of the cementitious matrixes depending on the type of fibre. Numerous fibre types have been employed, such as steel, polypropylene, carbon, sisal and glass fibres, among others. In the case of glass fibres, one of their main applications has been, in combination with cement mortar, forming glass fibre-reinforced cement, commonly termed GRC. The use of short randomly distributed glass fibres provides cement mortar with a remarkable improvement in certain mechanical properties. Not only is the tensile and flexural behaviour of GRC improved, but also its ductility. Consequently, GRC boasts a remarkable stiffness and compressive strength, which are provided by the cement mortar, and notable tensile and flexural behaviour, which is provided by the glass fibres [1].

There are other features of GRC that should be underlined, such as a notable fire resistance and an easy mouldability. Moreover, GRC elements do not require any steel reinforcement. Thus, such elements are resistant to corrosion and can be manufactured with reduced thicknesses (usually 10 mm thick), making GRC significantly lighter than an analogue reinforced concrete element. This compendium of advantages makes GRC competitive in a wide range of applications, such as cladding panels, pipes, refurbishment of buildings, sewer liners, tunnel claddings and acoustic barriers, among others [2,3,4,5,6].

The mentioned applications do not introduce significant stresses in the material, and GRC elements are mainly considered as non-structural. Although there are structural elements performed with GRC, such as roof sections or telecommunication towers [7], there are still doubts regarding the durability of the glass fibres when placed in an alkaline environment such as the cement mortar matrix [8]. Although alkali-resistant fibres are commonly employed in GRC production, reductions in the tensile strength and ductility have been reported [9].

Several approaches have been taken by researchers seeking to hamper the deterioration of the material properties. Some authors tried to cover glass fibres with a protection layer [10]. Nevertheless, the most common practice has been to add certain chemical substances to the cement mortar [11,12,13,14,15,16]. Recently, and due to the surge of nanoscale materials, several studies have been developed seeking to increase the mechanical properties of GRC by adding nanosilica to the cement mortar formulation [17,18,19].

In most of studies, the long-term mechanical properties of GRC were assessed after subjecting the material to an accelerated aging process [20,21]. The most common method is immersing GRC samples in hot water tanks at a determined temperature for a certain period of time. The equivalence between immersion times and natural exposure periods for non-modified GRC matrixes is generally accepted [22]. Even some theoretical models have been developed [23,24,25]. Nevertheless, such models and equivalences are usually dependent of the weather conditions of the site of the research [26]. In the case of GRC matrixes that include any addition, such equivalences still raise some uncertainties because the high temperatures of the water in the tanks might modify the reactions that develop during the aging process [27]. Based on the latter, there is great interest in assessing the evolution of the mechanical properties of GRC manufactured with certain chemical additions after a being subjected to natural weather conditions for a certain period of time.

In order to widen the knowledge in this field, an experimental campaign was carried out, seeking to obtain the mechanical properties of two formulations of GRC manufactured with chemical additions after five years subjected to natural weather conditions. Moreover, the mechanical properties of such GRC formulations, both young and naturally aged, were compared based on previous studies found in the literature. Thus, the evolution of such features was assessed. Moreover, with the help of a digital image correlation (DIC) analysis, the damage evolution and cracking patterns both of young and aged GRC formulations were described.

The significance of the research lies in the determination of the evolution of the mechanical properties of two GRC manufactured with modified cement mortar matrixes without the uncertainties that entail the use of an accelerated aging process. By avoiding any of those procedures, it is possible to obtain reliable results that might be of great importance for considering the potential use of these types of GRC mixes in structural elements.

## 2. Material Manufacturing, Natural Aging Conditions and Test Setup

In this study, two GRC formulations were used. A GRC formulation in which a thermally treated pure kaolin, commercially known as Metaver, was employed [28] (named as GRC-M). Metaver reacts with the free portlandite (calcium hydroxide) to form added calcium silicate hydrates. The other GRC formulation, called GRC-P, was manufactured with a white, amorphous, alumino-silicate addition called Powerpozz [29]. This addition reacts with calcium hydroxide to also form compounds of hydrated cement. The chemical composition of both additives can be seen in Table 1.

The mentioned additions were used complementary to the cement dosage being in both cases equivalent to a 25% of the cement weight. The relative weights of the formulation components can be seen in Table 2. Silica sand with a 100% passing of the 1.6 mm sieve was employed in the production process of both GRC formulations.

Table 2 shows that the water/cement ratios varied among the formulations seeking to reach the workability required in the production process. Following common practice, 38 mm-long alkali-resistant chopped glass fibres were added to the mixes in a 5% volumetric fraction [1].

Using the mentioned combinations, several 1.2 × 1.2 m^2^ test boards were produced by simultaneously spraying the cement mortar and the glass fibres directly into the moulds. All boards boasted an approximate thickness of 10 mm. Due to the reduced thickness, the orientation of the fibres could be considered almost orthotropic, with the fibres being almost parallel to the two main dimensions of the boards. After setting, the boards of GRC-M and GRC-P alike were stored in a climatic chamber at 20 °C and 95% of relative humidity until reaching 28 days of age.

A 50 mm-width frame was cut and discarded in all test boards in order to avoid testing bent fibres. Afterwards, 178 × 55 mm^2^ coupons were cut using a water-cooled circular saw. While some coupons were used in an experimental campaign for obtaining their mechanical properties at young ages, others were subjected to natural weather conditions for five years. The samples were placed in an open field facility, with all the sides of the samples except for one being subjected to sunlight, rainfall or any other weather condition. The exposure to natural weather occurred in Madrid (Spain). The temperature, relative humidity and rainfall can be seen in Figure 1.

After five years of natural exposure, the coupons were ground until a constant thickness was reached. Finally, a 3 mm-deep and 1 mm-wide notch was carried out in the central section of all coupons. The appearance of one of the coupons can be seen in Figure 2.

As GRC elements boast, in most cases, an approximate thickness of 10 mm and they do not include any type of reinforcement, assessing the evolution of the fracture energy with time is of key importance to evaluate the suitability of using GRC in structural elements with load-bearing capacity. The test setup was defined in accordance with a previous study where the fracture energy was assessed in young GRC [30]. Consequently, any variation of the materials properties could not have been caused by any change in the test setup. It should be noted that the test setup, although inspired in one of the most used recommendations for obtaining the fracture energy of plain concrete, did not maintain the dimensions of the samples, as they are not representative of GRC elements [31]. The tests were carried out in a universal testing machine equipped with a 1 kN load cell. Two linear differential transformers (LVDT) were placed at both sides of the tested sample to register its deflection. Complementarily to these gauges, a crack mouth open displacement strain gauge (CMOD) was attached to the lips of the notch with the help of two steel blades. In addition, the position of the actuator was also registered. The time and all the data previously mentioned were recorded with the help of a computer data acquisition system. The test was controlled by imposing the opening rate of the CMOD gauge—further details can be found in [30]. This rate was set as it enabled us to obtain a stable cracking process.

The lateral surface of the samples was painted in black and afterwards a speckled white pattern was carried out in order to perform the digital image correlation (DIC) analysis. The five-megapixel camera used for recording the cracking process could not be placed in front of the sample as one of the LVDT gauges was in this position. Thus, a mirror was placed between the LVDT gauge and the sample at 45° and the central zone of the sample where the notch was could be defined as the region of interest (ROI) in the DIC. A sketch of the ROI can be seen in Figure 3. Before performing the DIC analysis, a geometric calibration of the images acquired was performed in order to detect any possible distortions or inaccuracies.

## 3. Results and Discussion

### 3.1. Test Results

In Figure 4, the results obtained in the fracture tests performed in naturally aged GRC-M (samples manufactured with Metaver were called Me2, Me4 and Me5) coupons can be seen. The three curves plotted boast a similar shape. In this shape, a first loading branch with a notable stiffness and a linear behaviour started in the beginning of the test and ended with a progressive reduction in the stiffness near the limit of proportionality (LOP). Comparing the peak load of Me2, Me4 and Me5, it can be observed that in the case of Me2, the peak load was about 30 N higher than that registered in the other two samples. The second stretch boasted a similar slope in all samples to those obtained in the fracture tests of the GRC specimens [31]. In this stretch, the shape of the curves resembles an exponential softening function. It can be observed than Me2 was able to sustain greater load for the same values of deflection and CMOD throughout the test. A visual inspection of Me2 corroborated that there were no geometrical differences that could explain the observed variations in the mechanical behaviour.

The fracture behaviour of the three samples manufactured with GRC-P (samples manufactured with Powerpozz were called P3, P5 and P6) can be seen in Figure 5. Similarly to the case of GRC-M, there was no notable scattering between the results of the three samples, either in the peak load or in the unloading stretch of the curves. All samples reached a peak load close to 137 N with only slight variations around this value. However, in the unloading part of the curve of P3, a greater load bearing capacity was registered between 0.5 and 2 mm of CMOD.

### 3.2. Discussion of the Test Results and Comparison with the Fracture Behaviour of Young GRC

In order to perform a comparison of the experimental behaviour of the two types of GRC previously shown, the average fracture behaviour curve of each formulation was obtained. Such curves can be seen in Figure 6. In this figure, it can be clearly seen that there were no great differences in the peak load that both formulations were able to sustain. Moreover, there were no remarkable differences in the CMOD or deflection values where such peak loads were registered. Another feature that should be noted is that the unloading stretch of GRC-M has a slope which is noticeably steeper than that of GRC-P. Consequently, the load bearing capacity of GRC-P in the unloading branch is much greater than that of GRC-M for the same values of CMOD of deflection.

To perform an accurate comparison among the mechanical behaviours shown in Figure 6, the peak load and the fracture energy were obtained from the curves that appear in Figure 4 and Figure 5. These data can be seen in Table 3, together with the coefficient of variation of each parameter. The fracture energy was computed from the experimental data using Equation (1).
(1)$$G_{f}=\frac{W_{f}}{\Omega }=\frac{\int_{\delta_{0}}^{\delta_{f}}L\;d\delta }{\Omega }$$

In Equation (1), *G_f_* stands for the specific fracture energy, *W_f_* represents the fracture work performed during the test, Ω is the ligament area, *L* defines the load registered in the load cell, and *δ* is the displacement of the loading cylinder.

As can be seen in Table 3, GRC-M registered a greater scattering than that of GRC-P. It should be underlined that there is about 10 N of difference between the peak loads of GRC-M and GRC-P. Regarding the fracture energy computed, observing Table 3, it can be clearly perceived that the fracture energy of GRC-M is only 61% of the fracture energy obtained in the GRC-P tests.

In Figure 7, a comparison between the mechanical behaviour of young and naturally aged GRC-M samples is shown. It should be highlighted that the formulation of the young and aged GRC mixes was the same. Figure 7 reveals that there is hardly any difference between the experimental curves shown. It should be underlined that the unloading part of both curves is quite similar. Consequently, it seems that the improvement in the material behaviour provided by the presence of fibres does not suffer any remarkable reduction even after five years of natural exposure to the weather. However, it could be mentioned that there is a minor reduction in the peak load registered in the tests carried out in GRC-M aged samples with respect to the unaged material.

Figure 8 shows the experimental curves obtained in the fracture tests performed in aged GRC-P samples and also the curves obtained in [30] for the same formulation at a young age. As happened in GRC-M curves, there are no great differences in the behaviour obtained in the tests performed in aged and young GRC-P samples. The average value of the peak load registered for both types of samples shows no considerable differences. Furthermore, it can be observed that the initial part of the unloading stretch is remarkably similar in both young and aged samples. Nevertheless, it seems that there is a slight loss of load bearing capacity and ductility in the aged GRC-P samples from 1 mm of CMOD onwards. This phenomenon can be clearly appreciated in the load–deflection curves, where the behaviour of the young material is notably more ductile than the aged one.

In order to determine the change in the material properties caused by the exposition to the natural environment, a comparison between the values of the peak load and the fracture energy obtained in the young and aged tests is carried out in Figure 9. It should be mentioned that in this figure, the properties of the material at a young age are considered as the reference in each formulation separately. In the case of GRC-M, it can be seen that after five years of natural exposure, the peak load registered is slightly above 85% of that registered at a young age. It can be easily perceived that there is a very limited reduction in the fracture energy of aged GRC-M with respect to that obtained when the material was young. It should be mentioned that in order to avoid considering spurious data at the far tail of the curves, the fracture energy for both formulations was computed only until the unloading curve reached 20 N. In the case of GRC-P, it can be seen that there was a more reduced decrease in the peak load registered in the aged samples with respect to the young samples. However, when analysing the fracture energy, it was perceived that there was a greater decline than in the case of GRC-M being only 75% of that registered when at a young age.

Another point that deserves to be analysed is the applicability and representability of the aging mechanisms through the aging methods employed in the literature. As immersion in hot water tanks is accepted as the reference method, a literature search was performed in order to find any study that assessed the fracture energy of GRC after carrying out an accelerated aging process. Figure 10 shows the comparison of the results obtained by [32] and those determined in this study. It should be borne in mind that while in [32] the fracture tests were performed using a four-point bending test on unnotched samples, the present study employed a three-point bending test setup on notched samples. Moreover, the GRC formulations were not manufactured using the same chemical additions. Consequently, a direct comparison among the results of [32] and the ones of the present study could not be made. Nevertheless, certain conclusions could be derived regarding the accuracy of the aging method. If the results shown in the shaded area of Figure 10, which were obtained assuming a linear aging process, are compared with the results of this study, it can be perceived that the natural weather conditions of Madrid had a more limited deleterious effect on GRC-M than in the other formulations. On the contrary, GRC-P seems to have suffered a greater harm. In any case, it seems that the equivalence between the accelerated aging period by immersion in water at 80 °C and natural aging time shown in [32] may fit reasonably well.

## 4. Digital Image Correlation Analysis

In the past 20 years, there have been great advances in optical measurement methods which have proved to be versatile and accurate in a wide range of materials [33,34,35]. When applied to cementitious materials, DIC has been successfully applied to describe cracking caused by shrinkage, compressive loads [36,37] and flexural damage [38]. Moreover, in a previous publication, it was successfully applied to describe GRC progressive damage when subjecting it to fracture tests [32]. Based on the latter, it was decided to describe the damage patterns that appear in the naturally aged GRC formulations and also to compare them with the correspondent young GRC formulations.

In Figure 11, an image of the strain fields registered when the peak load was reached can be seen. Slight differences in the horizontal strain component of the two formulations analysed were detected. In order to provide a valuable comparison, the same colour scale values were employed. GRC-M and GRC-P strain maps had similar features. In both cases, the greatest strains were detected in the inner part of the spike that appears in the top of the notch. However, it can be seen that GRC-M has a wider area with a reddish distribution of strains than GRC-P. On the contrary, in GRC-P, this reddish zone is closer to the top of the sample than in the case of GRC-M. Moreover, it can be also perceived that GRC-P developed the cracking area at a certain angle with respect to the loading point that is right above the notch tip. The load–CMOD curve and the strain maps were compared, and it can be observed how the reduction in the stiffness of the material behaviour is related to the strain concentration of the red-coloured spike. Therefore, it might be said that in the red-coloured zone, the cracking process had already begun.

With the help of DIC, it was possible to obtain the strain fields of the naturally aged GRC samples. In order to assess the onset and evolution of cracks, the test was divided in ten stages: before starting the test, at 50% of the peak load, at the peak load and every 10% of reduction in the load with respect to the peak one until 30% of the peak load was reached. These stages were numbered from 1 to 10. It should be highlighted that the colour scale changes in the different images. In Figure 12, the stages regarding GRC-M can be seen. The first image illustrates the initial strain map without any deformations. In the image that shows the strain map at 50% of the maximum load, no load concentration in any part of the ROI can be perceived. When the peak load was reached in image 3, a red spike appeared at the top of the notch, crossing more than half of the ligament section. This reddish zone did not widen while the sample unloaded; on the contrary, it grew towards the loading cylinder as the test progressed.

Information regarding the damage evolution in young GRC-M can be seen in Figure 13. It should be mentioned that data from [30] were employed. In order to ease the comparison between the strain fields of young and aged GRC-M strain fields, the colour scale that appeared in [30] was modified. Image 2 does not reveal any remarkable strain concentration. Even at the peak load, there was not a clear damaged zone in the whereabouts of the tip of the notch. When 80% of the peak load was reached (image 5), a red-coloured zone could be clearly seen. This area was located between the tip of the notch and the middle part of the ligament. As the test progressed, this area widened without much progression towards the loading cylinder.

The procedure used for analysing the damage evolution of GRC-M was also applied to GRC-P. In Image 2 of Figure 14, it is possible to identify a small damaged area at the top of the notch. This area could clearly be perceived when the peak load was reached, with it being wider and larger than in the previous image. It should be pointed out that as the test progressed, the damaged area only grew towards the upper side of the sample, without any remarkable increment in the width of this area being observed. Another feature that could be mentioned is that the tip of the crack diverged from the path towards the loading point when approximately 60% of the peak load was reached. Later, when 30% of the peak load was reached, the crack direction changed, finally heading towards the loading point.

In the case of the analysis of the damage evolution of unaged GRC-P, the comments mentioned for unaged GRC-M could be applied. Observing Figure 15, it can be seen that when the peak load was reached, the damaged area is slightly greater than the corresponding area in Figure 14. It seems that the unaged GRC-M is capable of distributing damage to the sides of the damaged zone without requiring a notable growth in the length of the cracks towards the loading point.

## 5. Conclusions

A characterisation of the effect of natural aging in the fracture energy of two GRC formulations manufactured with Metaver and Powerpozz additions has been carried out. Moreover, a comparison with the performance of the same unaged GRC formulations, shown in [30], was accomplished. Based on the experimental results obtained, it can be concluded that the fracture energy of GRC-P after five years of exposure to natural environment is greater than the one of GRC-M. According to the comparison between the aged and unaged test results, it should be underlined that while the fracture energy of GRC-M only suffered a minor variation, while in the case of GRC-P, its fracture energy was reduced to 75% of the unaged material. Consequently, it seems that, if GRC elements with structural roles are desired, it would be advisable to employ GRC formulations similar to that of GRC-M. In any case, Powerpozz and Metaver alike seem to be suitable for reducing the decay of GRC fracture energy.

The data obtained in the experimental campaign of this study were compared with previous data from the literature [32], where several GRC formulations were aged by immersing the material in hot water. Based on such comparison, and assuming a linear degradation of the material over time, it seems that there are no remarkable differences between the evolution of the fracture energy obtained after a period of accelerated aging by immersion in hot water and the equivalent period of natural exposure. The slight discordances that appear might be caused by the differences in the formulation of the GRC matrixes or even in the test setup. In any case, further analysis in this matter would be of interest.

Based on the images obtained in the DIC analysis, it can be perceived that naturally aged GRC formulations were not capable of distributing damage in the area close to the tip of the notch. As the test progressed, cracks grew mostly towards the loading cylinder without any remarkable change in the width of the damaged area. On the contrary, young GRC seems to be locally more ductile as the width of the strain concentration spike grows as the test progresses, without showing notable crack growth towards the loading point. Moreover, if images of naturally aged GRC-M and GRC-P are compared, it can be seen that the damage evolution has a remarkable resemblance. Moreover, the two aged materials seem to distribute damage in a smaller area than when at a young age, and energy might have been consumed in the growth of the cracks towards the loading point.

## Figures and Tables

**Figure 1 materials-14-03355-f001:**
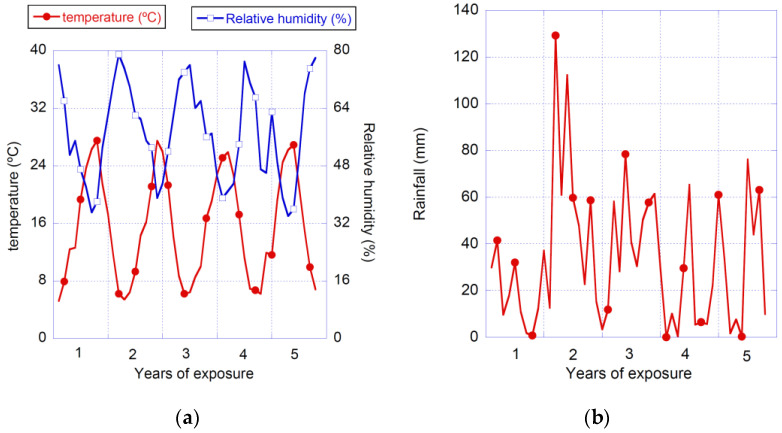
Climatic conditions of the exposure to natural weather (**a**) temperature and relative humidity (**b**) rainfall.

**Figure 2 materials-14-03355-f002:**
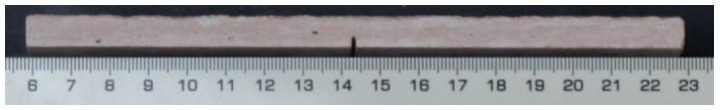
GRC coupon after preparation. Measures in cm.

**Figure 3 materials-14-03355-f003:**
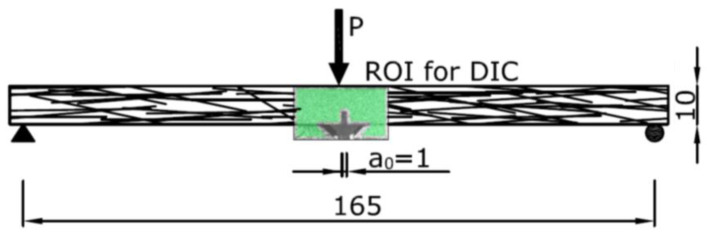
GRC test setup with the region of interest of the DIC highlighted in green. Measures in mm.

**Figure 4 materials-14-03355-f004:**
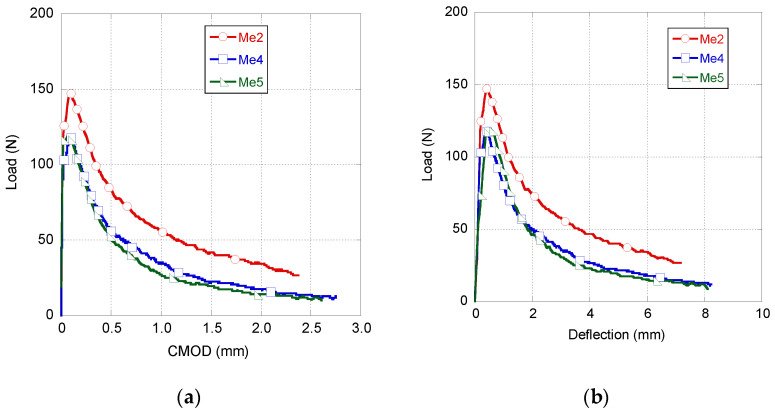
GRC-M fracture test results. Load–CMOD (**a**) and load–deflection (**b**) curves.

**Figure 5 materials-14-03355-f005:**
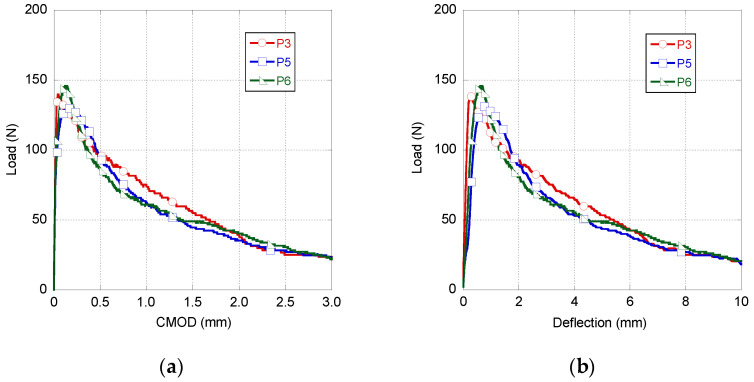
GRC-P fracture test results. Load–CMOD (**a**) and load–deflection (**b**) curves.

**Figure 6 materials-14-03355-f006:**
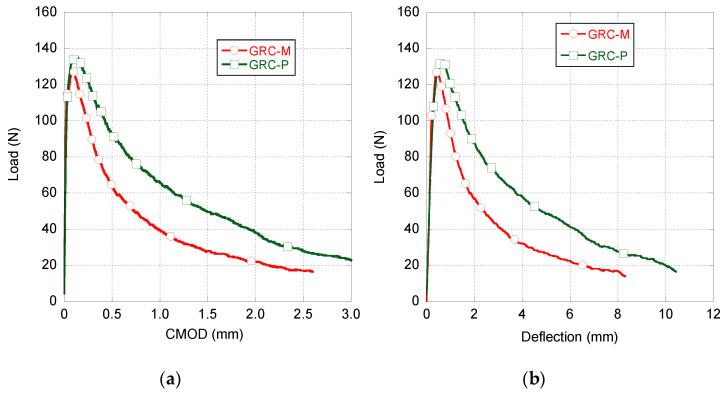
Average fracture behaviour of GRC, GRC-M and GRC naturally aged samples. Load–CMOD (**a**) and load–deflection (**b**) curves.

**Figure 7 materials-14-03355-f007:**
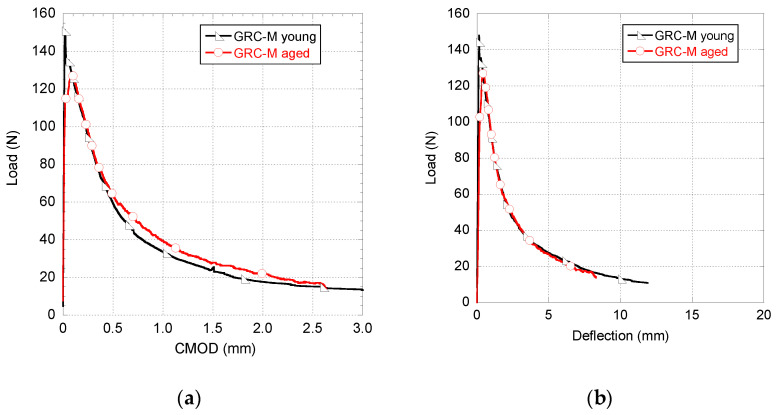
Comparison between the mechanical behaviour of young and naturally aged GRC-M samples. (**a**) Load–CMOD curve and (**b**) load–deflection curve.

**Figure 8 materials-14-03355-f008:**
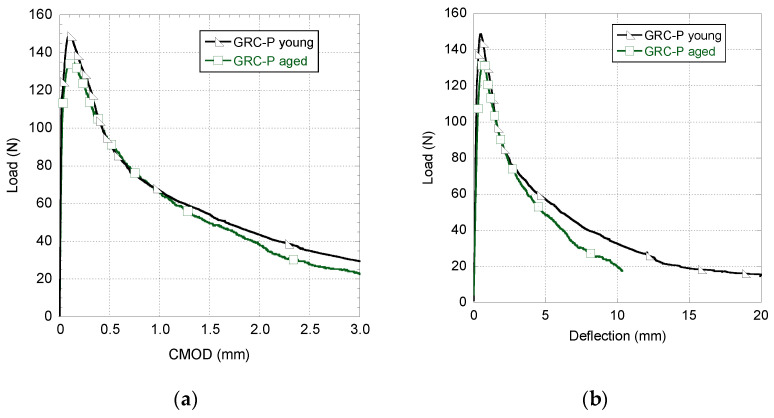
Comparison between the mechanical behaviour of young and naturally aged GRC-P samples. (**a**) Load–CMOD curve and (**b**) load–deflection curve.

**Figure 9 materials-14-03355-f009:**
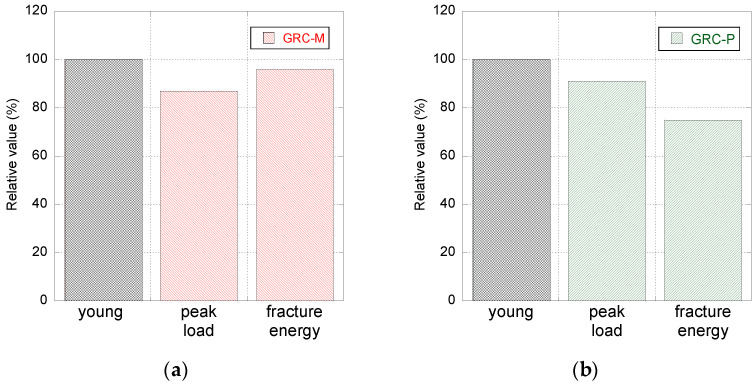
Evolution of peak load and the fracture energy of young and aged GRC-M (**a**) and GRC-P (**b**) expressed in relative terms. The material properties at young age are considered as the reference (100%).

**Figure 10 materials-14-03355-f010:**
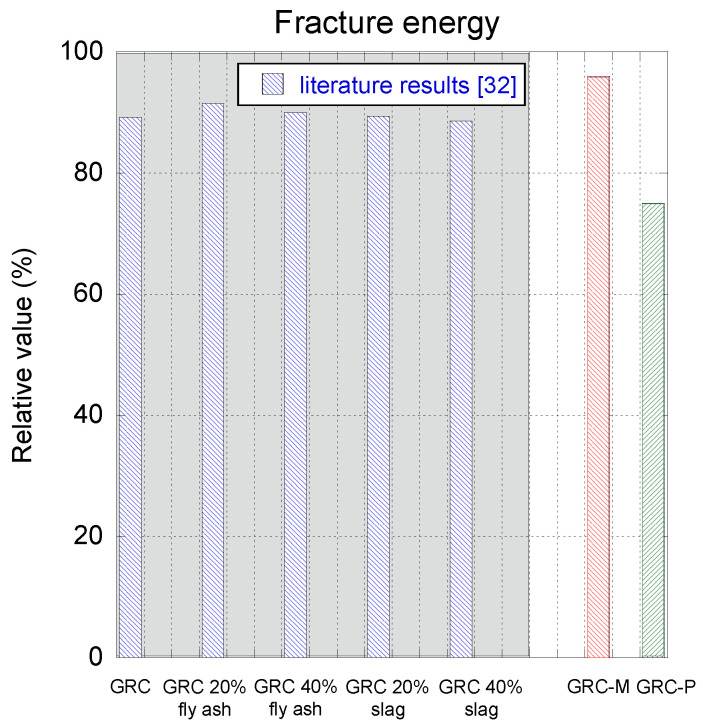
Comparison of the reduction in the fracture energy using accelerated aging methods [32] and natural weather exposure.

**Figure 11 materials-14-03355-f011:**
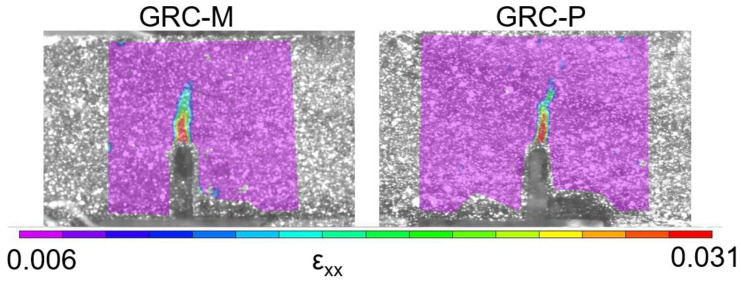
Strain distribution at maximum load.

**Figure 12 materials-14-03355-f012:**
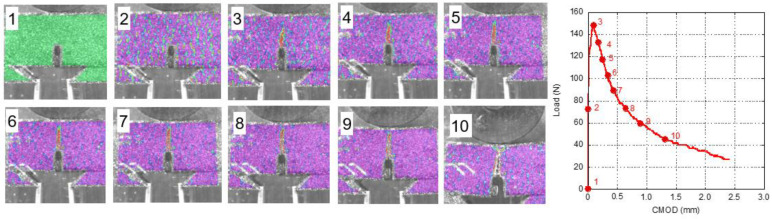
Strain field during the fracture test of a naturally aged GRC-M sample. Strain field is rescaled in each image.

**Figure 13 materials-14-03355-f013:**
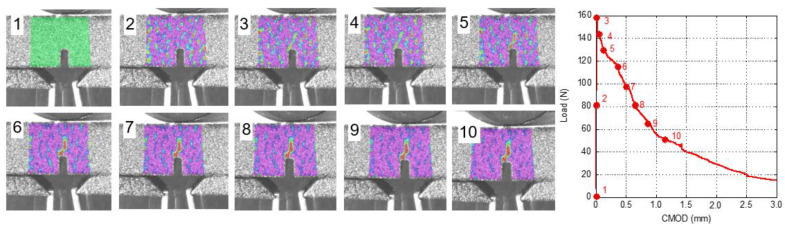
Strain field during the fracture test of a young GRC-M sample. Strain field is rescaled in each image [30].

**Figure 14 materials-14-03355-f014:**
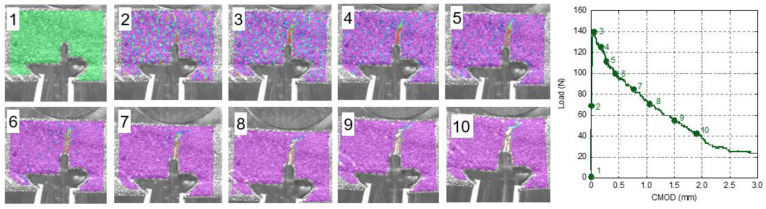
Strain field during the fracture test in a naturally aged GRC-P sample. The strain field is rescaled in each image.

**Figure 15 materials-14-03355-f015:**
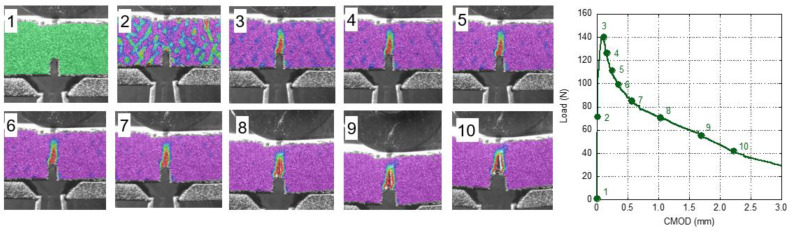
Strain field during the fracture test of a young GRC-P sample. Strain field is rescaled in each image [30].

**Table 1 materials-14-03355-t001:** Chemical composition of the commercial additives used.

	Powerpozz	Metaver
SiO_2_	52–55	52–54
Al_2_O_3_	41–44	40–42
Fe_2_O_3_	<1.90	<2.5
TiO_2_	<3	<1.0
SO_4_	<0.05	-
P_2_O_5_	<0.2	-
CaO	<0.2	<0.5
MgO	<0.1	<0.4
K_2_O	0.75	<2.0
Na_2_O	-	<0.1
LOI	<0.50	Not declared

**Table 2 materials-14-03355-t002:** Formulations of GRC manufactured expressed in relative weights.

	GRC-M	GRC-P
Cement	1	1
Addition	0.25	0.25
Sand	1	1
Water	0.46	0.5
water/cement	0.46	0.5
Plasticiser	0.01	0.01

**Table 3 materials-14-03355-t003:** Peak load and fracture energy of the naturally aged GRC formulations.

	GRC-M	GRC-P
Peak load (N)	128.29	138.90
c.v	0.13	0.05
Fracture energy (N/m)	959.29	1559.90
c.v.	0.28	0.05

## Data Availability

The data presented in this study are available on request from the corresponding author.

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
