# Peer review of "Influence of Natural Weather Conditions in the Long-Term Fracture Energy of Glass Fibre Reinforced Cement (GRC) Modified with Chemical Additions"

_materials, 2021, doi:10.3390/ma14123355_

Round 1
Reviewer 1 Report
The manuscript “materials-1250282”, entitled “Influence of natural weather conditions in the long-term fracture energy of Glass Fibre Reinforced Cement (GRC) modified with chemical additions”, discussed and evaluated the mechanical properties of two GRC produced with modified cement mortar matrixes and the digital imaging correlation analysis used for the damage evaluation and describing their cracking patterns. The manuscript is well-written, and discussions are sufficient. It is recommended to carefully check the manuscript to avoid some typos or grammatical errors. The manuscript should be accepted for publication after minor revision.
Some comments are as below:
- The quality of writing could be improved through careful checking of the manuscript to avoid typos.
For example:
- Line 95, Page 2:
“The chemical composition of both additions can be se seen in Table 1.”, it should be: both additives. Remove “se”.
- Line 113, Page 3:
at 20ºC, the sign for temperature must be corrected in the whole manuscript. It must be without underline.
- Line 134, Page 4:
…although inspired in one of one of the most…, “one of” is repeated.
- Line 159, Page 5:
Please define Me2, Me4 and Me5 in the manuscript.
- Line 162, Page 5:
…of Me2 and Me3 and Me5 it can… according to the manuscript, it should be: …Me2 and Me4 and Me5….
- Line 172, Page 5:
Please define P3, P5 and P6 in the manuscript.
- Line 247, Page 9:
In Figure 10, “literature results [32]”, according to the discussion in the manuscript, it should be: “literature results [35]”.

Author Response
Some comments are as below:
The quality of writing could be improved through careful checking of the manuscript to avoid typos.
For example:
- Line 95, Page 2:
“The chemical composition of both additions can be se seen in Table 1.”, it should be: both additives. Remove “se”.
The authors thank the reviewer for the comment. The typographic error has been corrected.
- Line 113, Page 3:
at 20ºC, the sign for temperature must be corrected in the whole manuscript. It must be without underline.
The authors thank the reviewer for the comment. The typographic error has been corrected.
- Line 134, Page 4:
…although inspired in one of one of the most…, “one of” is repeated.
The authors thank the reviewer for the comment. The typographic error has been corrected.
- Line 159, Page 5:
Please define Me2, Me4 and Me5 in the manuscript.
The authors thank the reviewer for the comment. The following sentence has been added to the manuscript to avoid any possible misunderstanding.
In Figure 4, the results obtained in the fracture tests performed in naturally aged GRC-M (samples manufactured with Metaver were called Me2, Me4 and Me5) coupons can be seen.
- Line 162, Page 5:
…of Me2 and Me3 and Me5 it can… according to the manuscript, it should be: …Me2 and Me4and Me5….
The authors thank the reviewer for the comment. The following wording has been added to the manuscript to avoid any possible misunderstanding.
Comparing the peak load of Me2 and Me4 and Me5 it can be observed that in the case of Me2 the peak load was about 30N higher than the one registered in the other two samples.
- Line 172, Page 5:
Please define P3, P5 and P6 in the manuscript.
The authors thank the reviewer for the comment. The following sentence has been added to the manuscript to avoid any possible misunderstanding.
The fracture behaviour of the three samples manufactured with GRC-P (samples manufactured with Powerpozz were called P3, P5 and P6) can be seen in Figure 5.
- Line 247, Page 9:
In Figure 10, “literature results [32]”, according to the discussion in the manuscript, it should be: “literature results [35]”.
The authors thank the reviewer for the comment. Reference [32] has been changed for reference [35]

Reviewer 2 Report
An experimental campaign seeking to assess the reduction of the fracture energy of two GRC manufactured with modified matrixes after five years of exposure to weather has been performed. Furthermore, a comparison with results of literature that used accelerated aging methods has been performed.
- pure kaolin, pozzolanic addition – where they came from and what pozzolana - natural, artificial.
- After setting, the GRC boards were stored in a climatic chamber at 20ºC and 95% of relative humidity until reaching 28 days of age. – it is about GRC-M and GRC-P?.
- Figure 8. Comparison between the mechanical behaviour of young and naturally aged GRC-P (a) load-CMOD curve (b) load-deflection curve – change colors.
- Figure 10. Comparison of the reduction of the fracture energy using accelerated aging methods [35] and natural weather exposure. – here is literature 35 and in figure 32. Which is correct.
- However, it can be seen that GRC-M has greater strains in the tip of the notch than GRC – GRC-P.
Author Response
An experimental campaign seeking to assess the reduction of the fracture energy of two GRC manufactured with modified matrixes after five years of exposure to weather has been performed. Furthermore, a comparison with results of literature that used accelerated aging methods has been performed.
- pure kaolin, pozzolanic addition – where they came from and what pozzolana - natural, artificial.
The authors appreciate the accurate comment of the reviewer. In order to clarify the text, a more detailed description of the products used has been included as follows:
In this study, two GRC formulations were used. A GRC formulation in which a thermal treated pure kaolin commercially known as Metaver was employed [30] (named as GRC-M). Metaver reacts with the free portlandite (calcium hydroxide) to form added calcium-silicate-hydrates. The other GRC formulation, called GRC-P, was manufactured with a white, amorphous, alumino-silicate addition called Powerpozz [31]. This addition reacts with calcium hydroxide to form also compounds of hydrated cement. The chemical composition of both additives can be seen in Table 1.
- After setting, the GRC boards were stored in a climatic chamber at 20ºC and 95% of relative humidity until reaching 28 days of age. – it is about GRC-M and GRC-P?.
The authors appreciate the accurate comment of the reviewer. The sentence has been modified and the following wording has been added.
After setting, the boards of GRC-M and GRC-P alike were stored in a climatic chamber at 20ºC and 95% of relative humidity until reaching 28 days of age.
- Figure 8. Comparison between the mechanical behaviour of young and naturally aged GRC-P (a) load-CMOD curve (b) load-deflection curve – change colors.
The authors thank the reviewer for the detailed revision of the manuscript. Figure 8 has been modified following the indications of the reviewer.
- Figure 10. Comparison of the reduction of the fracture energy using accelerated aging methods [35] and natural weather exposure. – here is literature 35 and in figure 32. Which is correct.
The authors thank the reviewer for the comment. Reference [32] has been changed for reference [35]
- However, it can be seen that GRC-M has greater strains in the tip of the notch than GRC – GRC-P.
The authors thank the reviewer for the comment. The following correction has been added to the manuscript to avoid any possible misunderstanding.
In both cases the greatest strains were detected in the inner part of the spike that appears in the top of the notch. However, it can be seen that GRC-M has a wider area with a reddish distribution of strains than GRC-P. On the contrary, in GRC-P such reddish zone is closer to the top of the sample than in the case of GRC-M. Moreover, it can be also perceived that GRC-P has developed the cracking area at a certain angle with respect to the loading point that is right above the notch tip.

Reviewer 3 Report
The manuscript deals about glass fibre reinforced cement (GRC) modified with chemical additives and the development of the fracture energy under natural aging. Therefore, an experimental program was conducted on notched GRC-samples, which had been exposed to natural weather conditions for five years. The manuscript is well organized and prepared in a good manner. However, I do have remarks, which should be incorporated before publication:
- Page 3, table 2: Please give information on the gravel size of the sand.
- Page 3, line 115-117: Please consider rephrasing of the sentences as it is confusing to the reader. What is meant with a frame that is cut “near” the edges? I guess you mean the “frame” was cut away at the edges. Also use cut instead of sewed.
- Page 3, line 118-119: Please provide more information on how the samples were stored and subjected to weather conditions. Have they been exposed to the weather conditions on all sides?
- Page 4, Fig 1a: replace Chinese letters in Figure by °
- Page 4, Fig.2: The resolution of the image is low and the numbers are almost not readable. Provide a better picture.
- Page 4, line 142: Replace aim by help
- Page 4, line 150-153: Can you proved a picture of the DIC measurement and the mirror. Have you checked for any inaccuracies with the DIC due to distortion of the measurement caused by the mirror?
- Page 5, line 159-160 and line 167-170: Please check for contradictory statements as in the beginning you mention that there was no remarkable scatter, while in the latter you explain that ME2 was able to sustain greater loads. Also replace than by that in line 167 before Me2
- Page 4, line 131-132 and page 7, line 208-209: Give information if the mix design was the same or if there have been any differences in the mix design between the naturally aged GRC and the young GRC taken from literature
- Page 8, Figure 8: Please use other colours or use a more light green, because in the current manuscript the green colour looks almost black.
- Page 9, line 250-267: Main critics here is why no own experiments on accelerated aging have been made on samples with the same mix design? It would not have caused that much effort but would provide a better comparison on the reliability of these tests.
- Page 10, line 283-284: I cannot see greater strains at the tip of the notch. Please make this more clear. However the strain field seems to have a bigger width in the case of GRC-M at first glance.
- Page 10, Figure 11: Change font size in figure to match font size of the text.
- Page 11, Figure 12 and 13: The strain measurements are barely recognisable. Increase the resolution of the subfigures and consider replacing the purple colour by a more brighter colour.
- Page 11, line 319: Reduced damage area in comparison to what? Please clarify.
Author Response
The manuscript deals about glass fibre reinforced cement (GRC) modified with chemical additives and the development of the fracture energy under natural aging. Therefore, an experimental program was conducted on notched GRC-samples, which had been exposed to natural weather conditions for five years. The manuscript is well organized and prepared in a good manner. However, I do have remarks, which should be incorporated before publication:
- Page 3, table 2: Please give information on the gravel size of the sand.
The authors thank the reviewer for the comment as the required information clearly improves the soundness of the contribution. The following sentence has been added to the manuscript.
The mentioned additives were used complementary to the cement dosage being in both cases equivalent to a 25% of the cement weight. The relative weights of the formulation components can be seen in Table 2. Silica sand with a 100% passing the 1.6mm sieve was employed in the production process of both GRC formulations.
- Page 3, line 115-117: Please consider rephrasing of the sentences as it is confusing to the reader. What is meant with a frame that is cut “near” the edges? I guess you mean the “frame” was cut away at the edges. Also use cut instead of sewed.
The authors would like to thank the reviewer as the remark pointed out that the comprehension of the sentence was compromised. Consequently, the authors have rewritten such sentence considering the modifications proposed by the reviewer. The final wording can be seen in the following lines.
A 50mm-width frame was cut and discarded in all test boards in order to avoid testing bent fibres. Afterwards, 178x55mm coupons were cut using a water-cooled circular saw.
- Page 3, line 118-119: Please provide more information on how the samples were stored and subjected to weather conditions. Have they been exposed to the weather conditions on all sides?
The authors thank the reviewer for its comment. The description of how the samples were stored and subjected to weather conditions have been improved. The final wording of such description can be seen in the following lines.
While some coupons were used in an experimental campaign for obtaining their mechanical properties at young ages, others were subjected to natural weather conditions for five years. The samples were placed in an open field facility being subjected to sunlight, rainfall or any other weather condition in all the sides of the samples except one. The exposure to natural weather occurred in Madrid (Spain).
- Page 4, Fig 1a: replace Chinese letters in Figure by °
The authors would like to point out that such characters appear due to a format character issue that is currently reported and will be solved in the final version of the contribution.
- Page 4, Fig.2: The resolution of the image is low and the numbers are almost not readable. Provide a better picture.
The authors thank the reviewer for her/his comment. The resolution of the image will be increased in the final version of the contribution.
- Page 4, line 142: Replace aim by help
The suggestion of the reviewer has been introduced in the final version of the text.
- Page 4, line 150-153: Can you proved a picture of the DIC measurement and the mirror. Have you checked for any inaccuracies with the DIC due to distortion of the measurement caused by the mirror?
The authors would like to point out that before performing any strain analysis a geometric distortion check was carried out in all the images acquired. In order to highlight such process in the contribution the following sentence has been added.
Thus, a mirror was placed between the LVDT gauge and the sample at 45º and the central zone of the sample where the notch is could be defined as the region of interest (ROI) in the DIC. A sketch of the ROI can be seen in Figure 3. Before performing the DIC analysis, a geometric calibration of the images acquired was performed in order to detect any possible distortion or inaccuracies.
- Page 5, line 159-160 and line 167-170: Please check for contradictory statements as in the beginning you mention that there was no remarkable scatter, while in the latter you explain that ME2 was able to sustain greater loads. Also replace than by that in line 167 before Me2
The authors would like to thank the reviewer for highlighting a possible contradiction in the manuscript. In order to clarify such issue, the wording has been modified as appears in the following lines.
The three curves plotted boast a similar shape. In such shape, a first loading branch with a notable stiffness and a linear behaviour started in the beginning of the test and ended with a progressive reduction of the stiffness near the limit of proportionality (LOP). Comparing the peak load of Me2, Me4 and Me5 it can be observed that in the case of Me2 the peak load was about 30N higher than the one registered in the other two samples.
- Page 4, line 131-132 and page 7, line 208-209: Give information if the mix design was the same or if there have been any differences in the mix design between the naturally aged GRC and the young GRC taken from literature
In order to clarify this comment, the following sentence has been added in the discussion of Figure 7 and Figure 8.
In Figure 7, a comparison between the mechanical behaviour of young and naturally aged GRC-M samples is shown. It should be highlighted that the formulation of the young and aged GRC mixes was the same. Figure 7 reveals (…)
- Page 8, Figure 8: Please use other colours or use a more light green, because in the current manuscript the green colour looks almost black.
The authors thank the reviewer for the comment and will change the colour pattern in the final version of the manuscript.
- Page 9, line 250-267: Main critics here is why no own experiments on accelerated aging have been made on samples with the same mix design? It would not have caused that much effort but would provide a better comparison on the reliability of these tests.
The authors agree with the comment performed by the reviewer. However, using an accelerated aging method at young GRC was not planned when the young GRC tests were performed (reference [34]).
It is true that when the tests of naturally aged GRC formulations were finished, it seemed as a valuable comparison adding the results of the young GRC coupons after being subjected to an accelerated aging process. Unfortunately, at this point the original test boards have been stored for five years in the climatic chamber and consequently if an accelerated aging was carried on such boards the comparison would not have been between young, artificially aged, and naturally aged GRC but between young, naturally aged and a mix between naturally and accelerated aged GRC. Consequently, the authors used test results found in literature to discuss the naturally aged results.
In any case, the authors thank the reviewer for the valuable suggestion and will consider it for future research.
- Page 10, line 283-284: I cannot see greater strains at the tip of the notch. Please make this more clear. However the strain field seems to have a bigger width in the case of GRC-M at first glance.
The authors have extended the explanation regarding Figure 11 in order to address the reviewer comment. The final wording can be seen in the following lines.
In both cases the greatest strains were detected in the inner part of the spike that appears in the top of the notch. However, it can be seen that GRC-M has a wider area with a reddish distribution of strains than GRC-P. On the contrary, in GRC-P such reddish zone is closer to the top of the sample than in the case of GRC-M. Moreover, it can be also perceived that GRC-P has developed the cracking area at a certain angle with respect to the loading point that is right above the notch tip.
- Page 10, Figure 11: Change font size in figure to match font size of the text.
The authors have changed the font sizes in Figure 11. Such figure can be seen in the its final form in the following lines.
- Page 11, Figure 12 and 13: The strain measurements are barely recognisable. Increase the resolution of the subfigures and consider replacing the purple colour by a more brighter colour.
The authors thank the reviewer for her/his comment. The resolution of the figures will be increased in the final version of the contribution.
- Page 11, line 319: Reduced damage area in comparison to what? Please clarify.
